# Public Health Implications of Wasting and Stunting Relationship in Children under Five Years Highly Vulnerable to Undernutrition in Guatemala: The REDAC Study

**DOI:** 10.3390/nu14193945

**Published:** 2022-09-23

**Authors:** Noemí López-Ejeda, Laura Medialdea, Antonio Vargas, Jessica Coronado, Miguel Ángel García-Arias, María Dolores Marrodán

**Affiliations:** 1Research Group EPINUT (Nutritional Epidemiology)—Unit of Physical Anthropology, Department of Biodiversity, Ecology and Evolution, Complutense University of Madrid, 28040 Madrid, Spain; 2Nutrition and Health Department, Action Against Hunger, 28002 Madrid, Spain; 3Nutrition and Health Department, Action Against Hunger, Guatemala City 01012, Guatemala

**Keywords:** acute malnutrition, chronic malnutrition, Composite Index of Anthropometric Failure (CIAF), weight-for-height z-score (WHZ), middle-upper arm circumference (MUAC)

## Abstract

(1) Background: Guatemala is the Latin American country with the highest prevalence of childhood stunting. Short height can bias the diagnosis of wasting when using the weight-for-height indicator. The aim of this study was to evaluate the diagnostic concordance of the anthropometric indicators of wasting and the relationship between wasting and stunting in children from highly vulnerable communities in Guatemala. (2) Methods: The sample consisted of 13,031 anthropometric records of children under five years of age (49.5% girls, average age of 27.9 months), including weight, height, and mid-upper arm circumference (MUAC), collected in March–August 2019. The proportions of stunting, underweight, and wasting, assessed by three different indicators, as well as their concurrence through the Composite Index of Anthropometric Failure were calculated. (3) Results: Stunting affected 73% of the sample, and 74.2% showed anthropometric failure. Wasting varied by indicator (weight-for-height: 2.8%; MUAC: 4.4%; MUAC-for-age: 10.6%). Concordance between MUAC and weight-for-height was very low (Kappa: 0.310; sensitivity: 40.9%). MUAC identified more wasted children in the stunted group (53.6% vs. 26.5%), while the opposite occurred in the non-stunted group (34.8% vs. 46.7%). (4) Conclusion: The presence of stunting affected the diagnosis of wasting, and both indicators should be included as diagnostic criteria for screening campaigns and in the treatment of moderate to acute wasting in vulnerable populations affected by multiple forms of undernutrition.

## 1. Introduction

The Central American Dry Corridor region has been the most affected historically, with a continuously high degree of food insecurity due to cyclical drought processes [1]. It is estimated that more than half of the small grain producers in Central America are based in such areas; they are subsistence farmers who harvest and consume the food they sow without the capacity to accumulate reserves, which makes them highly vulnerable to these adverse climatic events and endangers family subsistence [2]. In Guatemala, food insecurity also affects the western region, which has a complex geography: it is very mountainous and has vast relief, which hinders access to health services and other basic resources. 

In 2019, Guatemala was the second poorest country in Latin America, with 61.1% of its inhabitants living in multidimensional poverty [3]. The country is an extreme outlier in the region regarding chronic malnutrition, since almost half of the child population suffers from stunting [4]. This problem is even more significant in rural areas and indigenous groups [5].

Stunting is closely related to delayed neuromotor development. As stated by Guatemala’s government, children with stunted growth: learn to sit, stand, and walk later; have lower cognitive capacity; perform worse in school and are more likely to repeat academic years; miss more days of school due to illness; and are more likely to drop out than children with good nutritional status. It is estimated that, if trends do not change, by 2026, Guatemalan children will have missed almost 33 million equivalent years of learning in school and the country will lose Q146,207 million due to causes directly related to chronic malnutrition [6]. However, stunting does not imply an imminent risk to the lives of children who suffer it, unlike wasting, which requires immediate treatment due to the risk of rapid deterioration and death [7].

Wasting affects around 0.7% of the child population in Guatemala, but depending on the age group and geographical location, figures above 2% can be registered [5]. Currently, there is no gold standard criterion for diagnosing wasting in pediatrics. Children under five can be diagnosed indistinctly through two anthropometric indicators: the weight-for-height z-score (WHZ) or the mid-upper arm circumference (MUAC). Currently, the World Health Organization (WHO) recommends using both criteria for screening and admission for the treatment of severe and moderate cases [8]. However, the national protocol for epidemiological surveillance of wasting in Guatemala only contemplates WHZ or the presence of evident clinical signs (visible wasting or edema) [9]. Likewise, these are the only diagnostic criteria accepted for inclusion in treatment in nutritional recovery centers [10]. At the community level, MUAC is accepted as a criterion for inclusion in the outpatient treatment of severe cases [11], but not in moderate cases [12].

Research studies have repeatedly shown that both anthropometric indicators identify different population groups, and that the presence of short height could condition the diagnosis of acute malnutrition if the WHZ indicator is used [13]. Accordingly, this study aimed to evaluate the diagnostic concordance of both wasting indicators and the relationship between wasting and stunting in a sample of children under five years of age from highly vulnerable communities in Guatemala.

## 2. Materials and Methods

The REDAC study (from the Spanish “Relación entre Desnutrición Aguda y Crónica”) is a cross-sectional observational study funded by the European Civil Protection and Humanitarian Aid Operations Commission (ECHO) as part of a humanitarian food assistance intervention in highly food-insecure communities of Guatemala. The study involved eight humanitarian organizations performing anthropometric monitoring of vulnerable communities from March to August 2019 in 14 municipalities of four departments (Huehuetenango, Quiché, Chiquimula, and Baja Verapaz) selected for the impact of droughts in recent years on rural households’ food insecurity (Appendix A). The parents or guardians of the evaluated children signed informed consent for using their anthropometric data in the research. The data protection team and Ethics Committee of Action Against Hunger Spain ensured the protection and integrity of all participants.

### 2.1. Anthropometry and Nutritional Status Indicators

All of the anthropometrists attended a workshop on harmonization in measurement. The weight (kg), height (cm), and MUAC (mm) of children from 0–59 months were measured following the SMART guidelines for humanitarian interventions [14] with the same duly calibrated material (Appendix A). The children’s weight was taken using a SECA scale with an accuracy of 1 g, with the children wearing a t-shirt of their known weight. In the case of children unable to stand upright, the double-weighing method was performed, whereby the anthropometrist held the child in their arms and then subtracted their own weight recorded at the beginning of the same session. The height was taken using a wooden measuring rod provided by UNICEF that helps measure the length lying down in children unable to stand upright, in children under 24 months of age, children with a height of less than 87 cm, and the height standing upright in the rest of the children. MUAC was taken using a flexible plastic bracelet provided by Action Against Hunger with colors that determine the different degrees of severity of acute malnutrition. The measurement was taken in the medial area of the arm between the anthropometric points of the acromion on the shoulder and the radiale at the elbow.

From these direct measurements, the nutritional status of the participants was calculated based on various anthropometric indicators: stunting or chronic malnutrition was assessed through the height-for-age z-score (HAZ), underweight or global malnutrition through the weight-for-age z-score (WAZ), and wasting or acute malnutrition through the presence of edema or three different anthropometric indicators (WHZ, MUAC-for-age z-score (MUACAZ), and MUAC) as an absolute value with fixed cut-off points (moderate wasting < 125 mm and severe wasting < 115 mm). For HAZ, WAZ, WHZ, and MUACAZ, deviation values were calculated with respect to the World Health Organization Child Growth Standards [15] through the Anthro software [16] (z-score for moderate undernutrition between −2 and −3 and for severe undernutrition < −3).

The Composite Index of Anthropometric Failure (CIAF) [17] was applied to measure the joint presence of several types of undernutrition. Anthropometric failure is defined as the single or combined presence of any type of undernutrition diagnosed by simple anthropometric measurements (weight, height, and arm circumference). Accordingly, it is possible to account for the proportion of individuals without anthropometric failure (those not presenting stunting, wasting, or underweight).

### 2.2. Data Quality Assurance

The consolidated database initially included 13,124 records. A total of 91 records of children over 59 months and another without sex were eliminated, leaving 13,031 records in the final database. Of those, 49.5% (6447) were girls, and the average age was 27.9 (14.5–42.5) months. Appendix A shows the final composition of the sample by geographic area, sex, and age. Of the 13,031 final records, only 16 had no weight registration (0.13%) and 24 had no record of height (0.18%), although cases without MUAC data were more numerous with 1169 missing records (9.0%). Most of this missing MUAC data corresponds to children under six months of age, a period for which there is currently no treatment for acute malnutrition, and its diagnosis is recommended by WAZ or WHZ. However, the rest of the anthropometric data of these children were retained for analysis.

A strict data cleaning protocol was applied based on the WHO–UNICEF recommendations for anthropometric surveys [18]: (a) fixed exclusions: z-score fixed limits above which the measure is not compatible with life; (b) flexible exclusions: calculated according to the samples’ distribution by eliminating all values deviating ±4 SD from the median. Appendix A shows the cleaning process results for each anthropometric indicator.

### 2.3. Statistical Analysis

The analysis was performed using IBM SPSS v.25 software (Endicott, New York, NY, USA). All tests were carried out under a bilateral hypothesis, considering a significance level of *p* < 0.05. For continuous variables, the normality was assessed using the Kolmogorov–Smirnoff test with Lilliefors correction. Accordingly, Student’s *t*-test or the Mann–Whitney test was used to compare central parameters. Contingency tables were used to compare proportions, and the chi-square test with Yates correction was applied. 

Univariate logistic regression analysis was applied to quantify the association between the undernutrition categories through the odds ratio (OR). Cohen’s Kappa test was used to analyze the agreement between anthropometric indicators by considering the concordance thresholds proposed by Landis and Koch [19]. Using WHZ as the reference classification, the following were calculated: sensitivity, or the probability of correctly classifying a wasted child (true positive/(true positive + false negative)); specificity, or the probability of correctly classifying a non-wasted child (true negative/(true negative + false positive)); accuracy, or the ability to correctly diagnose wasted and non-wasted children ((true positive + true negative)/total); negative likelihood ratio, or the probability of obtaining a negative result when a child suffers from wasting ((1 − sensitivity)/specificity).

## 3. Results

Table 1 summarizes the results of the anthropometric assessment by sex. Stunting was the most prevalent nutritional problem, affecting around three-quarters of the sample. Except for underweight (WAZ), significant differences were found between sexes for the other types of undernutrition. When wasting was evaluated by WHZ and MUACAZ, boys were more affected, whereas girls were more affected if MUAC was used. Furthermore, the proportion of wasting showed significant differences when comparing the indicators two by two (*p* < 0.001), with WHZ revealing the lowest figures. Regarding the detection of severe wasting cases, WHZ and MUAC detected the same proportion of cases, while MUACAZ identified more than twice as many children.

Table 2 shows that the proportion of individuals without anthropometric failure was very low (26%). The presence of stunting and its combination with underweight are the two most common problems. As for the comparison by sex, boys accumulated a significantly higher percentage of anthropometric failure and stunting. Additionally, there were 146 (1.1%) concurrent cases of stunting and wasting diagnosed by WHZ and/or MUAC, representing 20.8% of the total wasting cases. There were also six cases of edema, which is indicative of severe wasting (0.05%).

The concordance analysis (Appendix A) showed that MUAC identified only 40.9% of the cases diagnosed by WHZ. Similarly, only 28.3% of those identified by MUAC were also identified by WHZ. These results imply that there is a low concordance index (Κ = 0.310; asymptotic error: 0.021; *p* < 0.001). Analyzing the diagnostic efficacy of MUAC, the results were 40.9% sensitivity, 96.9% specificity, 95.2% accuracy, and 61.0% likelihood ratio. 

Table 3 shows the results of the relationship between acute and chronic malnutrition. Regardless of the indicator, the probability of being wasted was significantly higher among the stunted group, with OR values between 1.6 and 4.3 (higher in girls). The wasting indicator that showed the greatest differences was MUACAZ, followed by MUAC. 

Figure 1 shows the proportion of cases that would enter treatment according to the WHO-recognized criteria. MUAC identified more cases in the total sample and among those with stunting. The opposite result was found in children without stunting, where WHZ identified more cases. The proportion of children diagnosed by the two methods simultaneously was similar in the presence or absence of stunting. Performing this analysis by sex, a significantly different pattern was revealed (*p* < 0.001) because, in boys, WHZ and MUAC diagnosed a similar proportion of cases (WHZ: 38.2%; MUAC: 37.7%; both: 24.1%), while in the female series, MUAC identified many more cases of wasting (WHZ: 20.1%; MUAC: 63.7%; both: 16.2%).

## 4. Discussion

The results confirm the nutritional vulnerability of this sample by finding the prevalence of undernutrition to be markedly higher than the national figures reported in the last National Survey of Maternal and Child Health [5]: 47% of infant stunting has been reported at the national level, with 16.6% being severe cases. In the present study, those figures reached 73.0% and 39.3%, respectively. Regarding the underweight cases, the national prevalence was 12.6%, with 2.1% of severe cases compared to 36.3% and 11.4% in the current study. Concerning wasting, the national survey only contemplates cases assessed by WHZ, which results in a prevalence of 0.7% with 0.1% of severe cases. In our study, we obtained 2.8% and 0.6%, respectively, when applying the same criteria. However, using the other anthropometric indicators, those figures were notably higher (4.4% and 0.6% through absolute MUAC and 10.6% and 1.4% through MUACAZ). 

Likewise, 74.2% of the children showed some anthropometric failure. This value is markedly higher than the 51.3% reported for an international study with more than 100,000 children from 14 countries [20]. Another large-scale study with 39 countries also reported lower figures (47.5% in 1990–2000 and 42.6% in 2001–2014) [21]. In that study, the CIAF was between 27% and 31% higher among mothers with primary education compared to those with higher levels and 21% higher among people in the lowest wealth quartile compared to the highest.

Anthropometric failure was significantly higher in boys, coinciding with the results of the meta-analysis published by Thurstants et al. [22] This phenomenon has been described as female ecostability, which argues that girls generally have lower body sizes and this require less energy for maintenance. They also have increased adiposity, making them more resistant to a lack of food or recurrent infections [23]. However, in the case of wasting diagnosed by MUAC, the prevalence was higher for girls (5.0% vs. 3.7%), while the opposite result was obtained with WHZ (2.1% vs. 3.4%).

In the present study, 1% of cases with the three types of simultaneous undernutrition were recorded (67 boys and 53 girls). Another study with anthropometric surveys from 51 countries showed that the concurrence of all types of undernutrition implies a 12.3-fold increased risk of death versus a normal nourished state, whereas that risk value is 2.3 for those children with wasting only and 1.5 for those with stunting only [24]. Based on this evidence, an international movement advocates for rethinking the traditional anthropometric categorization used for admission and discharge in the therapeutic and supplementary feeding programs [25].

The present study confirmed the low concordance between WHZ and MUAC, which shows a prevalence of wasting of 2.8% and 4.4%, respectively. In another Guatemalan survey with 906 vulnerable children from Zona Reina (Quiché), a lower prevalence of wasting by WHZ (1.8% vs. 2.5%) was also found [26]. Research studies have shown that the two indicators, which were indistinctly used for wasting diagnosis and access to treatment, identify different population groups and are associated differently with the risk of death [27]. The relationship between WHZ and MUAC varies markedly between countries, as demonstrated by Grellety and Golden [28]. In the case of Guatemala, their study showed that, of the total number of children with wasting, MUAC alone identified 70.1% of the cases compared to just 13.0% identified by WHZ alone and 16.9% by both simultaneously. In our study with highly vulnerable children, these differences were less marked (51%, 29%, and 20%, respectively). 

To explain the population differences in the discrepancy between MUAC and WHZ, it can be thought that MUAC could better reflect wasting in populations with a high prevalence of stunting, which is an indicator that is independent of height. Our study showed that MUAC and MUACAZ identify a higher proportion of children with wasting in the stunted group than WHZ (4.8%, 11.9%, and 2.9%, respectively). In contrast, these differences are not as evident in children without stunting (1.9%, 3.2%, and 1.8%). In addition, of all wasting cases identified in the stunted group, the majority were based on MUAC alone (53.6%), while in the non-stunted group, the highest proportion was identified by WHZ alone (46.7%). A similar trend was found in the aforementioned international study (20), where the prevalence of wasting was markedly higher in the group with stunting using MUAC (15.2% vs. 5.9% without stunting), and only slight differences were found using WHZ (13.3% vs. 11.5%). That study reported an overall MUAC–WHZ concordance (K: 0.358) similar to that of the present study (0.310). However, they found that the agreement was higher in the stunted group (K: 0.486 vs. 0.292 without stunting). 

However, the highest prevalence of acute malnutrition was obtained with MUACAZ, which would reach figures within the high emergency threshold according to the current classification of the WHO (10–15%) [29]. This is consistent with what is reported in a study with 882 nationally representative surveys from 41 different countries, where MUACAZ was the indicator that reported the highest figures [30]. In that study, it was also shown that the use of MUACAZ slightly improved the agreement with WHZ compared to the unadjusted MUAC. Another study found that MUAC and MUACAZ have the same predictive ability for short-term mortality risk between 6 and 35 months [31]. There is controversy as to which wasting indicator reflects the greatest risk of death [27,32,33], and there are population differences that may also be related to the concurrence of stunting [24].

There are pilot experiences in Guatemala with children over five years of age testing the use of MUAC bracelets adapted to introduce age adjustment for the classification of nutritional status [34]. Their qualitative analysis showed that 71% of the anthropometrists preferred the MUACAZ bracelet to weight and height assessment. However, only 62.5% indicated that the MUACAZ bracelet was either easy or very easy to use. Given that MUAC is much easier to use at the community level than MUACAZ, public health researchers recommend the traditional MUAC as a preference indicator for acute malnutrition in screening campaigns and nutrition rehabilitation programs because the limited improvements of MUACAZ do not justify the added complexity [30,31].

A limitation of the present study is that it included a convenience sample not representative of the territory or the general population, given that the participating communities were selected based on vulnerability criteria. On the contrary, it was an adequate sample for the study’s objective, given that the undernutrition risk, specifically wasting risk, was higher than in the general Guatemalan child population.

## 5. Conclusions

The study results highlight the need to include MUAC and WHZ as diagnostic criteria for wasting in vulnerable populations with a high prevalence of stunting. This should be considered for epidemiological surveillance, community screening campaigns, and access to therapeutic or nutritional supplementation programs for moderate and acute wasting. Currently, MUAC is not included as a valid criterion in the national surveillance protocols. In the present study, 358 cases with acute malnutrition by MUAC were recorded, which would not have been identified by WHZ and would not have access to the necessary treatment. Additionally, MUAC is accepted for screening in the community, but if a child is identified with moderate wasting by MUAC, it should be confirmed by WHZ at the treatment site. The present study confirmed the low concordance between those indicators and, therefore, a child identified in the community may not be admitted to outpatient treatment. This situation endangers the children’s lives and can cause community disaffection with the nutrition programs committing their attendance in the future. 

However, the inclusion of MUAC as a diagnostic criterion may lead to an increase in the prevalence of acute malnutrition among the most vulnerable populations and, therefore, may require more resources and personnel for managing cases at the community level.

## Figures and Tables

**Figure 1 nutrients-14-03945-f001:**
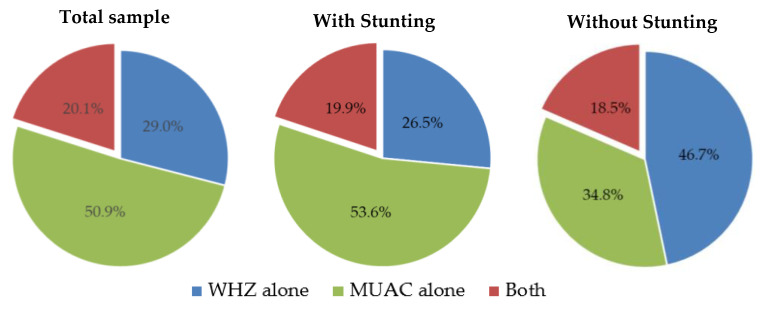
Proportion of wasting diagnosed by weight-for-height (WHZ) and/or mid-upper arm circumference (MUAC) as a function of stunting.

**Table 1 nutrients-14-03945-t001:** Anthropometric indicators and prevalence of the different types of undernutrition by sex.

Anthropometric Indicators	Total Median [IQR]	Boys Median [IQR]	Girls Median [IQR]	*p*-Value
Weight (kg)	10.0 [8.1, 12.3]	10.2 [8.2, 12.4]	9.9 [7.9, 12.1]	<0.001
Height (cm)	79.4 [70.3, 87.7]	79.9 [70.5, 87.7]	79.0 [10.2, 87.7]	0.028
MUAC (mm)	144 [135, 153]	145 [137, 153]	143 [135, 152]	<0.001
HAZ (z-score)	−2.7 [−3.4, −1.9]	−2.8 [−3.5, −2.0]	−2.7 [−3.4, −1.9]	<0.001
WAZ (z-score)	−1.6 [−2.4, −0.91]	−1.7 [−2.4, −0.9]	−1.6 [−2.4, −0.9]	0.147 ^NS^
WHZ (z-score)	−0.07 [−0.7, 0.6]	−0.09 [−0.8, 0.6]	−0.05 [−0.7, 0.6]	0.020
MUACAZ (z-score)	−0.8 [−1.4, −0.2]	−0.8 [−1.5, −0.13]	−0.8 [−1.4, −0.2]	0.482 ^NS^
**Malnutrition** **Categories**	**Total** **% (N)**	**Boys** **% (N)**	**Girls** **% (N)**	***p*-value**
STUNTING (HAZ)	73.0% (9119)	74.9% (25.1)	71.0% (4387)	<0.001
Moderate	33.7% (4211)	33.6% (2125)	33.8% (2086)	0.877 ^NS^
Severe	39.3% (4908)	41.3% (2607)	37.2% (2301)	<0.001
WASTING (WHZ) *	2.8% (355)	3.4% (222)	2.1% (133)	<0.001
Moderate	2.2% (278)	2.7% (175)	1.6% (103)	<0.001
Severe	0.6% (77)	0.7% (47)	0.5% (30)	0.064 ^NS^
WASTING (MUAC) *	4.4% (519)	3.7% (223)	5.0% (296)	0.001
Moderate	3.8% (452)	3.3% (196)	4.3% (256)	0.003
Severe	0.6% (67)	0.5% (27)	0.7% (40)	0.102 ^NS^
WASTING (MUACAZ) *	10.6% (1253)	11.1% (662)	10.0% (591)	0.051 ^NS^
Moderate	9.2% (1091)	9.7% (716)	8.8% (517)	0.098 ^NS^
Severe	1.4% (162)	1.5% (889)	1.3% (74)	0.294 ^NS^
UNDERWEIGHT (WAZ)	36.3% (4678)	37.1 (2413)	35.5% (2265)	0.053 ^NS^
Moderate	24.9% (3206)	26.1% (1697)	23.6% (1509)	0.001
Severe	11.4% (1472)	11.0% (716)	11.8% (756)	0.139 ^NS^

HAZ: height-for-age z-score; IQR: interquartile range; MUAC: mid-upper arm circumference; MUACAZ: MUAC-for-age z-score; ^NS^: not-significant (*p* > 0.05); WAZ: weight-for-age z-score; WHZ: weight-for-height z-score. * The prevalence of acute malnutrition differs significantly across all indicators (WHZ vs. MUAC, WHZ vs. MUACAZ, and MUAC vs. MUACAZ) and for all categories (global, moderate, or severe) with a significance level of *p* < 0.001.

**Table 2 nutrients-14-03945-t002:** Composite Index of Anthropometric Failure (CIAF) compared by sex.

	Total % (N)	Boys % (N)	Girls % (N)	*p*-Value
Without failure	25.8% (3220)	23.8% (1503)	27.8% (1717)	<0.001
* Wasting only	0.3% (39)	0.2% (14)	0.4% (25)	0.067 ^NS^
* Wasting + Underweight	0.1% (18)	0.2% (13)	0.1% (5)	0.066 ^NS^
Stunting + * Wasting + Underweight	1.0 (120)	1.1% (67)	0.9% (53)	0.244 ^NS^
Stunting + Underweight	32.8% (4092)	33.4% (2108)	32.2% (1984)	0.127 ^NS^
Stunting only	39.2% (4886)	40.4% (2545)	38.0% (2341)	0.005
Underweight only	0.8% (95)	0.8% (52)	0.7% (43)	0.411 ^NS^
CIAF	74.2% (9250)	76.1% (4799)	72.3% (4451)	<0.001

* Wasting diagnosed by weight-for-height (<−2 z-score) and/or mid-upper arm circumference (<125 mm); ^NS^: Not significant (*p* > 0.05).

**Table 3 nutrients-14-03945-t003:** Prevalence and probability of wasting as a function of the existence of stunting by sex.

	Wasting Indicator	Total Sample	With Stunting	Without Stunting	Difference	ꭓ^2^ *p*-Value	Odds Ratio	95% C.I.
Total	WHZ	2.8% (355)	2.9% (268)	1.8% (62)	+1.1%	0.001	1.621	1.226–2.143
	MUAC	4.4% (519)	4.8% (420)	1.9% (52)	+2.9%	<0.001	2.614	1.953–3.499
	MUACAZ	10.6% (1253)	11.9% (1032)	3.2% (82)	+8.7%	<0.001	4.337	3.448–5.454
Boys	WHZ	3.4% (222)	3.5% (166)	2.3% (36)	+1.2%	0.015	1.566	1.089–2.255
	MUAC	3.7% (223)	3.9% (173)	1.7% (21)	+2.2%	<0.001	2.380	1.507–3.760
	MUACAZ	11.1% (662)	12.3% (550)	3.9% (49)	+8.4%	<0.001	3.462	2.566–4.669
Girls	WHZ	2.1% (133)	2.3% (102)	1.5% (26)	+0.8%	0.028	1.619	1.049–2.499
	MUAC	5.0% (296)	5.9% (247)	2.1% (31)	+3.8%	<0.001	2.879	1.972–4.203
	MUACAZ	10.0% (591)	11.5% (482)	2.3% (33)	+9.2%	<0.001	5.589	3.907–7.994

C.I.: confidence interval; MUAC: mid-upper arm circumference; MUACAZ: MUAC-for-age z-score; WHZ: weight-for-height z-score.

## Data Availability

The data presented in this study could be available on request from the corresponding author. The data are not publicly available due to they come from a consortium of different organizations with different data protection and data sharing policies.

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
