# Peer review of "Public Health Implications of Wasting and Stunting Relationship in Children under Five Years Highly Vulnerable to Undernutrition in Guatemala: The REDAC Study"

_nutrients, 2022, doi:10.3390/nu14193945_

Round 1

Reviewer 1 Report

This study tested the influence of stunting on the definition of wasting ascertained by WHZ +/- MUAC in food-insecure Guatemala. The findings may be important for public health.

I have a few comments:

1. "The data protection team and Ethics Committee of Action Against Hunger Spain always ensured the protection and integrity of all participants." Please, state whether this ethics committee has approved the study. Is this an accredited medical research ethics committee (MREC)? How could a Spanish ethichs committee approve a study that is performed in another country (with different legislation)?

2. Anthropometric indicators are easy to use in practice for diagnosing wasting, but what is the gold standard? Please, elaborate already in the Introduction.

3. Please, explain throughout the manuscript what anthropometric failure actually means.

4. The English should be improved.

Author Response

Dear Reviewer,

We thank you for your time and effort in reviewing our article and for your comments that allow us to improve its quality.

Below we respond to each of your comments and have highlighted the changes in the manuscript in yellow.

Best regards,

This study tested the influence of stunting on the definition of wasting ascertained by WHZ +/- MUAC in food-insecure Guatemala. The findings may be important for public health.

I have a few comments:

  1. "The data protection team and Ethics Committee of Action Against Hunger Spain always ensured the protection and integrity of all participants." Please, state whether this ethics committee has approved the study. Is this an accredited medical research ethics committee (MREC)? How could a Spanish ethichs committee approve a study that is performed in another country (with different legislation)?

The study has not passed any medical research committee since it is an observational study and does not involve any intervention or invasive procedure (only the three basic anthropometric measurements that are usually taken in screening and malnutrition prevention campaigns are taken).

As mentioned in the methodology, the study is part of a humanitarian assistance project, which, like all Action Against Hunger projects, has been monitored by its own regulatory mechanisms, those from the donor (the European Civil Protection and Humanitarian Aid Operations Commission, ECHO) and subjected to possible external audits. Within these procedures is always the collection of informed consent from parents or guardians for the measurements of their children.

The Action Against Hunger network is organized by central headquarters coordinating country missions. In this case, the Guatemala mission depends directly on the Spanish headquarters and is subject to the country's own regulatory mechanisms and those of the Spanish headquarters, which is where the Ethics Committee and the data protection and auditing department are located.

  1. Anthropometric indicators are easy to use in practice for diagnosing wasting, but what is the gold standard? Please, elaborate already in the Introduction.

In general, when talking about child malnutrition, there is a clear separation between the diagnosis made in socio-economically developed contexts and impoverished contexts. In the first case, malnutrition is usually associated with hospitalization for another disease, where diagnostic resources may be greater. However, as Joosten and Hults point out in a review, even in this context, there is no consensus on which diagnostic criterion is the gold standard for pediatric ages [1].

In the case of low-resource settings, anthropometry is the tool universally used and is recommended and endorsed by the WHO (as is already mentioned in the introduction section when we said “the World Health Organization (WHO) recommends using both criteria for screening and admission to the treatment of severe and moderate cases”). When searching for malnutrition on the WHO website, only basic anthropometric measurements are mentioned as a diagnostic criterion (weight-for-age, height-for-age, weight-for-height…) [2]. Within the anthropometric criteria, WHZ is usually considered as the gold standard for wasting diagnosis between 6 and 59 months but it is demonstrated that depending on the population, MUAC can be a better mortality predictor.

We have added the appointment that there is no clear gold standard for wasting in pediatrics in the introduction section and clarification about mortality risk prediction in the discussion section supported for several references.

  1. Joosten and Hulst. Nutritional screening tools for hospitalized children: Methodological considerations. Clinical Nutrition, 2014; 33 (1): 1-5. https://www.sciencedirect.com/science/article/pii/S0261561413002070?casa_token=J45s-zZkgxcAAAAA:brKwM4uJ4lLxhVXB9SgtG07Qf58MePG785F5fLrSAq140yCBhaNi6tHM09z8Epo8s0rT9WYp
  2. World Health Organization (WHO). Malnutrition Fact Sheets. https://www.who.int/news-room/fact-sheets/detail/malnutrition
  3. Please, explain throughout the manuscript what anthropometric failure actually means.

This clarification has been added at the end of section 2.1.

  1. The English should be improved. The English writing has been revised by MDPI's own editing service. Attached is the certificate that endorses it.

Reviewer 2 Report

Thank you for conducting this analysis! Comparison of several measures of wasting in a vulnerable population has implications well beyond Guatemala. This reviewer recommends  some minor changes:

line 28: ".. stunting conditions the diagnosis of.. The word conditions is hard to understand.. please replace this word or the entire sentence for better understanding

ine36: Please move the phrase: " with a continuously high degree of food insecurity" to after the word "historically" in the same sentence change 

line 40: change to read: "...it also affects the western region"..

line 70: replace "recurrently" with the word "repeatedly"

Author Response

Dear Reviewer,

We thank you for your time and effort in reviewing our article and for your comments that allow us to improve its quality.

We have corrected all the errors appointed. All changes have been highlighted in green in the manuscript. In addition, the English writing has been revised by MDPI's own editing service (Attached is the certificate that endorses it).

Best regards,
